# Association between Quantitative Electroencephalogram Frequency Composition and Post-Surgical Evolution in Pharmacoresistant Temporal Lobe Epilepsy Patients

**DOI:** 10.3390/bs9030023

**Published:** 2019-03-04

**Authors:** Raúl Roberto Valdés Sedeño, Lilia María Morales Chacón, Abel Sánchez Coroneux

**Affiliations:** 1Hospital Clínico Quirúrgico Hermanos Ameijeiras, Havana 10400, Cuba; rrvaldess@infomed.sld.cu; 2Centro Internacional de Rehabilitación Neurológica, Havana 10400, Cuba; abel@neuro.ciren.cu

**Keywords:** temporal lobe epilepsy, surgery, quantitative electroencephalogram, post-surgical evolution

## Abstract

The purpose of this paper is to estimate the association between quantitative electroencephalogram frequency composition (QEEGC) and post-surgical evolution in patients with pharmacoresistant temporal lobe epilepsy (TLE) and to evaluate the predictive value of QEEGC before and after surgery. A prospective, longitudinal study was made at International Neurological Restoration Center, Havana, Cuba. Twenty-nine patients with TLE submitted to epilepsy surgery were evaluated before surgery, and six months and two years after. They were classified as unsatisfactory and satisfactory post-surgical clinical evolution using the Modified Engels Scale. Eighty-seven electroencephalograms with quantitative narrow- and broad-band measures were analyzed. A Mann Whitney test (*p* > 0.05) showed that QEEGC before surgery was similar between groups independently of two years post-surgical evolution. A Mann Whitney test (*p* ˂ 0.05) showed that subjects with two years satisfactory post-surgical evolution had greater alpha power compared to subjects with two years unsatisfactory post-surgical evolution that showed greater theta power. A Wilcoxon test (*p* ˂ 0.05) showed that alpha and theta power increased for two groups from pre-surgical state to post-surgical state. Logit regression (*p* ˂ 0.05) showed that six months after surgery, quantitative electroencephalogram frequency value with the greatest power at occipital regions shows predictive value for two years evolution. QEEGC can be a tool to predict the outcome of epilepsy surgery.

## 1. Introduction

Temporal lobe epilepsy (TLE) is the prototype syndrome that can be surgically treated. Several authors have described the prognostic value of clinical, electroencephalographic (EEG), and magnetic resonance imaging (MRI) variables for post-surgical evolution [1,2]. Most of the research has been focused on the predictive value of epileptiform discharges for post-surgical evolution, and while some results have shown strong predictive values of bad evolution [2,3,4,5,6,7,8,9], others have not found any relationship [10,11]. On the other hand, other authors have studied epileptic discharges and slow interictal activity without finding any relation between slow activity and seizure recurrence [12]. Nevertheless, it has been found that focal slow activity and epileptic discharges have been related to seizure recurrence during the post-surgical phase, mostly in patients with mesial TLE [8]. Still, predictive value has been widely studied and post-surgery predictive variables are yet to be found [13,14,15,16,17]. Despite of these investigations, there is no agreement concerning biomarkers of post-surgical evolution in this kind of patient. The EEGis a useful tool, innocuous and easily recording fluctuations of voltage in time. EEG abnormalities range from epileptiform discharges to slow activity, and variation of adult’s fundamental rhythms such as alpha and beta [18]. There are no studies about the predictive value of QEEGC in patients with TLE for seizure recurrence after surgery. We intend to study QEEGC in subjects with satisfactory and unsatisfactory post-surgical evolution, and to apply QEEGC as a predictor for seizure recurrence. 

## 2. Materials and Methods

A prospective, longitudinal study with intervention and cases was carried out at the International Restoration Center (CIREN), Havana, Cuba, between 2012 and 2015. Epileptic patients with pharmacoresistant focal epilepsy were referred from different regions of the country arrived from specialized clinics all over the country to the CIREN epilepsy surgery program once they showed no response to pharmacologic treatment. A standard pre-surgical protocol was applied to define the epileptogenic zone [19]. Subjects and EEG recordings were evaluated by two epileptologists (LM and RV) before surgery, and six months and two years after. Family and patient´s informed consent was received in all cases [19]. Data were collected prospectively.

### 2.1. Inclusion Criteria

Patients with TLE and surgical criteria were included according to guidelines and expert criteria of the institution, i.e., non-response to at least 2 appropriate antiepileptic drugs (AEDs) trials due to inefficacy and intolerance, hence recurrently compromised by seizures. Subjects submitted to TLE resection with follow-up for one year after surgery were included, meanwhile those with prior brain surgery were not. Patients older than 18 years were included after their written consent to participate in the study. EEGs were registered before surgery, and six months and two years after [20].

### 2.2. Exclusion Criteria

Progressive Systemic Diseases of Central Nervous System. Acute disorders non-seizure related. Lack of cooperation with pharmacologic treatment. Poor quality of electroencephalogram due to artifacts.

### 2.3. Ethical Considerations

All the procedures followed the rules of the Declaration of Helsinki of 1975 for human research, and the study was approved by the scientific and ethics committee from the International Center for Neurological Restoration (CIREN37/2012) [20]. 

### 2.4. Post-Operative Follow-Up

Post-surgical seizure outcome assessment was based on the system proposed by Engel. Engel class I, free of disabling seizures; class IA (completely seizure-free); class II, rare seizures (fewer than three seizures per year); class III, worthwhile improvement (reduction in seizures of 80% or more); class IV, no benefit [21]. For statistical analysis, class I was classified as “satisfactory” outcome, while classes II, III and IV as “unsatisfactory” seizure relief outcome [20]. Most used drugs were Carbamazepine, Valproic Acid, and Benzodiazepines; most frequent combination was Carbamazepine with Benzodiazepines and Valproic Acid with Benzodiazepines. Patients kept their AEDs for at least 2 years after surgery, without modification [20]. 

### 2.5. Surgical Procedure and Resection Size

Surgery consisted of anteromedial temporal lobectomy tailored by Electrocorticography (ECoG) recording. ECoG data acquisition was performed with a Medicid-5 digital EEG system (Neuronic SA, Havana, Cuba), using AD-TECH subdural electrodes (grid and strips) [20]. Sequential intraoperative Electrocorticography was monitored until epileptiform activity disappeared.

### 2.6. Histopathological Examination

Samples were taken in mesial and neocortical regions. Hippocampal sclerosis (HS) was defined by neuronal loss in CA1, CA3, and CA4 regions of the hippocampus. For focal cortical dysplasia (FCD) classification, the system proposed by the International League Against epilepsy was used. For Central Nervous System tumor histopathological diagnosis purposes, WHO classification was used [20,22,23]. Dual pathology was related to pylocitic astrocytom and arachnoid cystic in two of the patients [20].

### 2.7. Data Analysis

Digital Scalp EEG recording with duration of 30–60 minutes was reviewed and interpreted by two experienced neurophysiologist (LM) and (RV), using EEG Edition 6.3.1.1 software (Neuronic SA, Havana, Cuba), to assess QEEGC. Subjects were awake with their eyes closed, signal was recorded with 19 surface electrodes, with electrode reference between Cz-Pz, and filter 1–30 Hz (12 dB/oct.). Impedance was kept bellow 5 KOhm. Neuronic Medicid 5 equipment was used for EEG recording (Neuronic SA, Havana, Cuba). Windows of 2.56 seconds of minimum duration, free of artifacts were selected. 

Neuronic EEG Tomographic Quantitative 6.2.2.0 (Neuronic SA, Havana, Cuba) was used for quantitative analyses of narrow and broad frequency bands with eyes closed. For broad-band analysis, Absolute power (AP), Relative power (RP) and Medium frequency (MF) in each frequency band (alpha, beta, theta, delta, and total) were taken into account. Spectral analysis of narrow-band was made for 48 frequency points with spectral resolution of 0.390625 Hz, frequency bands (Delta 1.56–3.52 Hz, Theta 3.91–7.42 Hz, Alpha 7.81–12.50 Hz, and Beta 12.89–19.14 Hz). 

Statistica 8.0 was used for statistical processing, comparing age at surgery, age of first seizure, epilepsy duration, and quantitative EEG frequency composition between groups (StatSoft, Inc. (2007). STATISTICA (data analysis software system), version 8.0, www.statsoft.com.Tulsa, USA). A Mann Whitney test for independent samples and a Wilcoxon test for dependent samples were applied. Statistical significance was set at *p* < 0.05. Logit regression was used for post-surgical evolution prediction. Exact *p* values generated for small to moderate samples were taken for significance evaluation. 

To see if QEEGC was a major definer in patients with satisfactory and unsatisfactory post-surgical evolution in the second year, we compared groups attending to ipsilateral electrodes and contralateral electrodes to the epileptogenic zone. Thus, right and left epileptogenic zone were analyzed together, looking for changes in the surgical side and outside the surgical side. To calculate differences in pre-surgical phase and variations from pre-surgical to post-surgical phase, we followed the same procedure; patients were rearranged attending to the evolution achieved in the second year. Using narrow-band, the value of QEEGC with maximum energy in occipital regions was analyzed looking for predictive power of evolution in the second year.

## 3. Results

### 3.1. Demographic Profile and Pathology

Results of demographic profile are shown in Table 1. Age at surgery showed a mean of 34.37 ± 7.61, epilepsy duration a mean of 22.31 ± 10.61 and age of first seizure a mean of 12.76 ± 9.48. The difference between satisfactory and unsatisfactory post-surgical evolution groups was not statistically significant. Distribution of gender showed 16 female (55%) and 13 male (45%). In our sample, the most common etiology was FCD associated with a principal lesion (FCD type III) [20].

### 3.2. Narrow- and Broad-Band QEEGC in Pre-surgical Patients with Pharmacoresistance TLE

Patients with satisfactory and unsatisfactory post-surgical evolution in the second year showed similar frequency composition in pre-surgical phase. Figure 1a,b shows comparisons of alpha AP (*p* = 0.633) and theta AP *p* = (0.368) in pre-surgical phase. Narrow-band analysis and the rest of broad-band measures show the same behavior. There were no significance differences (Mann Whitney); therefore, this data was no use as predictor of evolution after the surgical intervention.

### 3.3. Narrow- and Broad-Band QEEGC Changes from Pre-surgical to Post-surgical Phase in Patients with Pharmacoresistance TLE

Table 2 show variations in EEG power from pre-surgical to post-surgical phase according to post-surgical evolution achieved in the second year. Only significant electrodes are shown; evaluation of all electrodes can be found in Appendix A (Table A1). There was an augmentation of alpha AP in satisfactory post-surgical evolution group in the second year that showed significance in frontal inferior ipsilateral electrode *p* = 0.017 and temporal anterior ipsilateral electrode *p* = 0.017 to the epileptogenic zone. Patients with unsatisfactory post-surgical evolution in the second year also had significant alpha AP augmentation in central ipsilateral electrodes *p* = 0.028 and temporal anterior ipsilateral *p* = 0.010 to the epileptogenic zone. Theta AP increased in satisfactory post-surgical evolution group in the second year in temporal anterior ipsilateral electrode to the epileptogenic zone *p* = 0.042, and in the same electrode *p* = 0.007 in unsatisfactory post-surgical evolution group.

Mean alpha absolute power augmentation in the second year was different for satisfactory and unsatisfactory post-surgical evolution group respectively (12.1 ± 4.4 µv^2^ Hz; 6.1 ± 1.7 µv^2^ Hz). Using mean comparison, these values showed significance differences, being greater in satisfactory post-surgical evolution group (*p* = 0.001). Mean theta absolute power augmentation in the second year was only slightly different for satisfactory and unsatisfactory post-surgical evolution group, respectively (8.9 ± 7.1 µv2Hz; 8.1 ± 3.1µv2Hz). Narrow-band analysis showed similar results and the rest of broad-band measures showed no significance data.

### 3.4. Narrow- and Broad-Band QEEGC Differences in Post-surgical Patients with Pharmacoresistance TLE

Figure 2 shows differences for satisfactory and unsatisfactory post-surgical evolution groups in the second year, considering ipsilateral and contralateral electrodes to the epileptogenic zone. Theta RP was significantly greater for patients with unsatisfactory post-surgical evolution group in the second year in most electrodes, except forfrontal inferior ipsilateral, frontal inferior contralateral, temporal anterior contralateral to epileptogenic zone and mid-central. Theta mean frequency in same patients showed significance in mid-line central *p* = 0.043 and mid-line frontal *p* = 0.034, parietal contralateral *p* = 0.005 and central contralateral *p* = 0.039, central ipsilateral *p* = 0.004, frontal superior contralateral *p* = 0.048 and frontal superior ipsilateral *p* = 0.031 electrodes. Compared to unsatisfactory post-surgical evolution group, satisfactory post-surgical evolution group had higher alpha AP in mid-Parietal *p* < 0.022, mid-frontal *p* < 0.043, temporal posterior contralateral *p* < 0.033 and ipsilateral *p* < 0.004, frontal inferior ipsilateral *p* < 0.006, occipital ipsilateral *p* < 0.025, frontal superior contralateral *p* < 0.025, front-polar contralateral *p* < 0.007 and ipsilateral *p* < 0.003 to epileptogenic zone. The same comparison for alpha RP found that all electrodes showed significance for satisfactory post-surgical evolution group.

Narrow EEG frequency analysis showed significance differences for satisfactory and unsatisfactory post-surgical evolution groups in the second year. The satisfactory group showed a significant increment in alpha frequency at 9.76, 10.15, 10.54, 10.93, 11.32 Hz with *p* < 0.02, unsatisfactory group showed a significance increase in low and high theta frequency values at (5, 5.46, 5.85, 6.25, 6.6, 7, 7.42 Hz) with *p* < 0.01. The same results were observed for post-surgical six-month analysis. Figure 3 is a representation of dissimilar findings of alpha and theta frequency measures in patients with satisfactory and unsatisfactory post-surgical evolution in the second year. Only significant measures are represented. A mirror effect can be observed: while satisfactory post-surgical evolution patients in the second year showed higher power of alpha measures, unsatisfactory post-surgical evolution patients showed higher power of theta measures.

### 3.5. Predictive Value of QEEGC After Surgery for Seizure Recurrence in Pharmacoresistance TLE Patients

The predictive value of QEEGC after surgery for two years seizure recurrence was analyzed; Table 3 shows that the frequency value with maximum energy in occipital regions in the first six months after surgery was a predictor of evolution in the second year. It is shown that the modeled probability that evolution is equal to satisfactory. Total of correct and incorrect prediction of cases for each group—satisfactory (13 correct and 3 incorrect) and unsatisfactory (6 correct and 6 incorrect)—is shown. While satisfactory evolution in the second year showed 81.25% of correct prediction, unsatisfactory evolution was only 50% correct. Interpretations of these results are made later in this paper.

Figure 4 and Figure 5 shows narrow-band quantitative analysis in patients with epilepsy in the right and left temporal lobe six months after surgery and different post-surgical evolution two years after. For satisfactory post-surgical evolution patients, alpha power was higher.

## 4. Discussion

The most important findings in this study are that QEEGC was different between subjects with satisfactory and unsatisfactory post-surgical evolution in the second year, and that at six months QEEGC showed predictive value for two-year seizure recurrence. Subjects that presented satisfactory post-surgical evolution in the second year showed greater values of alpha absolute and alpha RP measures compared to those showing unsatisfactory post-surgical evolution. The same subjects showed an increment of alpha and theta AP in post-surgical phase compare to pre-surgical phase. On the other hand, the observed changes in unsatisfactory post-surgical evolution group in the second year were greater theta relative power and theta mean frequency compared to satisfactory post-surgical evolution group. These subjects also presented augmentation of theta and alpha absolute power in post-surgical phase compared to pre-surgical phase. There were no significant differences in pre-surgical comparisons between satisfactory and unsatisfactory post-surgical evolution patients.

We have not found investigations comparing quantitative alpha power in patients with TLE before and after surgery, considering post-surgical evolution. There are several investigations of the predictive value after surgical intervention of clinical and electroencephalographic variables for seizure recurrence. Early seizure recurrence after surgery has been associated with bad prognosis [2,24,25]. Epileptiform activity after surgery has shown in some studies strong predictive value for seizure recurrence [2,3,4,5,6,7,8]. However, results are still controversial [10,11]. Most of the research has been focused on epileptiform discharges, while other pieces have included epileptiform discharges and slow interictal abnormalities [8,12]. Interictal slowing may reflect the epileptogenic process [26,27]. Presently, the controversy about the relative contributions of ictal scalp video EEG, routine scalp outpatient interictal EEG, intracranial EEG and MRI for predicting seizure-free outcomes after temporal lobectomy remains [20,28]. 

Several neural sources contribute to the generation of EEG functioning at high and low frequencies (0.5–30 Hz) [18]. Quantitative analysis decomposes the signal, allowing the study of dominant and non-dominant frequency bands, being more acute in the study of EEG [18]. Taking into account the findings of alpha power augmentation in satisfactory evolution group, many authors have considered mesial TLEas a network disease using EEG, positron emission tomography with fluorodeoxyglucose (PTE-FDG) and MRI, which affects not only the hippocampus but also neocortical temporal structures, extra-temporal, and subcortical-like basal ganglia and thalamus [29,30,31,32,33,34,35]. It has been found in patients with TLE atrophy of white and gray matter in thalamus [36,37,38]. Thalamus atrophy is frequent in TLE patients, being present in around 61% of ipsilateral patients and in 50% bilateral patients; its frequency is close to hippocampus atrophy (82% of cases) [39,40]. 

Goldman et al. (2002) studied the relationship between thalamus and alpha power using functional MRI in humans. They found that augmentation of alpha power correlated with an increment of blood oxygen level-dependent signal (BOLD) in the thalamus and insula [41,42]. Although generators of alpha rhythm are still in discussion, the interaction between thalamus and cortex is accepted [18]. In our opinion, subjects with satisfactory post-surgical evolution in the second year showed QEEGC composition closer to physiological parameters compared to unsatisfactory post-surgical evolution group. The resection of the epileptogenic zone and absence of seizures could provide better interaction of cortical networks and thalamus, improving neural functioning. 

Taking into account the augmentation of theta relative power and theta mean frequency in subjects presenting unsatisfactory post-surgical evolution in the second year, regional slowing of EEG has been associated with tumors and stroke [43,44,45]. The EEG of a healthy awake adult is composed of fast rhythms (8–30 Hz), whereas slow ones (0.3–7 Hz) are normally seen in functional states such as sleep and childhood [18]. Di Genaro et al. investigated the roll of interictal slow activity in epileptic patients operated due to mesial temporal sclerosis, tumors, and FCD. They did not find significant associations between seizure persistence and interictal slow activity, despite being present in post-surgical phase in 92% of patients in the first month, and 87% in the first year [12]. However, they did not apply quantitative analysis of EEG.

Several studies have shown the relationship between hypo-metabolism and slow activity in TLE patients, using PET-FDG. This could represent microscopic changes such as neuronal loss and synaptic reorganization of the temporal lobe [30,46,47,48]. In our sample, the most common etiology was FCD associated with a principal lesion (FCD type III); in recent years FCD has been identified as a major cause of pharmacoresistant TLE in young patients undergoing surgical resection [20,49,50]. James X. Tao et al. applied electro-corticography, EEG, and image studies in patient candidates to surgery, analyzing Interictal Regional Delta Activity (IRDA). They found that for neocortical and mesial TLE, it was a mixture of monomorphic and polymorphic delta-theta and epileptogenic activity. They conclude that IRDA was an epileptogenic marker of TLE, representing a network disease in mesial temporal epilepsy, involving neocortical structures [27]. This indicated that this disease affects structures beyond the mesial temporal lobe. It has been seen that in 20% of subjects that receives surgery the number of seizures will diminish keeping medication, 10% will remain the same and another 10% will get worse [51]. The unsatisfactory evolution group presented mostly simple focal seizure, with several of this group evolving to generalized tonic-clonic seizures.

Variations from pre-surgical to post-surgical phase showed the augmentation of alpha and theta AP in post-surgical phase for satisfactory and unsatisfactory evolution groups. However, behavior between both was different. We observed that satisfactory post-surgical evolution group had greater augmentation of alpha power over theta power, while unsatisfactory post-surgical evolution group had greater augmentation of theta power over alpha power. This could indicate, for the satisfactory post-surgical evolution group, some restoration of adult band frequency composition. Resection of epileptogenic zone could influence in posterior alpha rhythm, due to better functioning of neural networks. Moreover, if we consider that the elevation of alpha power was significantly greater for satisfactory post-surgical evolution group compared to unsatisfactory evolution one, in our opinion, patients that showed unsatisfactory post-surgical evolution in the second year had a lesser physiologic frequency composition compared to those with satisfactory post-surgical evolution. Seizure persistence could cause dys-functioning of cortical and thalamus–cortical interactions, justifying slow findings and alpha power decrement.

We found that by using quantitative EEG the value of frequency with maximum energy in occipital regions in the first six months after surgery showed predictive value for two-year evolution. We have not found investigations taking into account the predictive value of EEG frequency composition for post-surgical evolution. However, there are papers that have found slow changes in temporal and extra-temporal areas, classifying this entity like a network disease. Considering these results, Patrick et al. found that in patients surgically treated because of low-grade tumors and mesial temporal epilepsy, slow wave activity and epileptiform alterations were associated with seizure persistence. This association was greater for mesial temporal epilepsy [8]. We are aware that AED increases theta power and affects electrical activity [52]. In our sample, AED remained the same before and after surgical intervention, concerning doses or type and during the time of this investigation. We observed that the correct prediction of satisfactory post-surgical evolution group was higher (81.25%), compared to only 50% while predicting unsatisfactory evolution, Logit test *p* = 0.04. Changes from pre-surgical to post-surgical phase showed that for both groups alpha AP and theta AP augmented, though alpha increment was significantly higher (*p* < 0.05) in satisfactory evolution, while theta augmentation was not different between groups (Table 2). Consequently, analysis in the second year showed for unsatisfactory evolution that theta RP was significantly higher, which is the quotient between theta AP and the rest of absolute broad-band measures [53]. Thus, differences in AP after surgery were mainly related to alpha band. Taking into account these findings and the significance of the Logit test, this could indicate that an increment of alpha power after surgery is more accurate for evolution prediction than theta augmentation. However, our sample is small. Because of these limitations, future investigations may be required to understand the influence of these results. 

## 5. Conclusions

QEEGC can be a tool to predict the outcome of epilepsy surgery.

## Figures and Tables

**Figure 1 behavsci-09-00023-f001:**
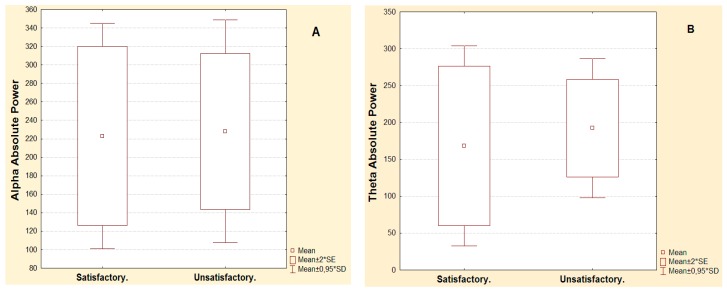
Patients in pre-surgical phase; regroup according to the evolution achieved in the second year of surgical intervention. (**A**) Alpha absolute power. (**B**) Theta absolute power. Mann Whitney.

**Figure 2 behavsci-09-00023-f002:**
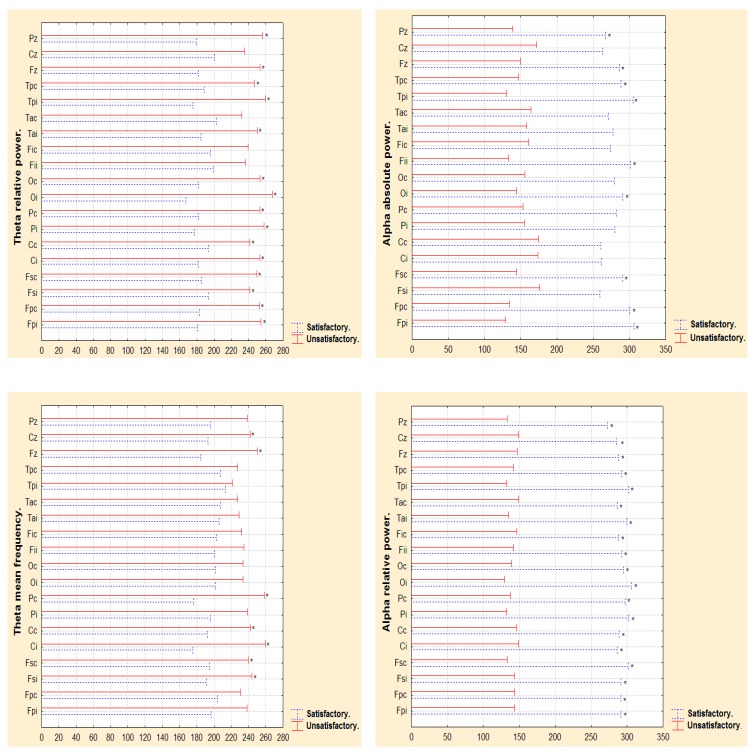
Mann Whitney. Differences in theta relative power and theta mean frequency, and alpha absolute power and alpha relative power, between patients with satisfactory and unsatisfactory post-surgical evolution in the second year after surgery. Fpc, Fsc, Cc, Pc, Oc, Fic, Tac, Tpc (Front-polar Contralateral, frontal superior Contralateral, central Contralateral, parietal Contralateral, occipital Contralateral, frontal inferior Contralateral, temporal anterior Contralateral, temporal posterior Contralateral electrodes), Fpi, Fsi, Ci, Pi, Oi, Fii, Tai, Tpi (Front-polar Ipsilateral, frontal superior Ipsilateral, central Ipsilateral, parietal Ipsilateral, occipital Ipsilateral, frontal inferior Ipsilateral, temporal anterior Ipsilateral, temporal posterior Ipsilateral electrodes), Fz, Cz, Pz (mid-line frontal, mid-line central, mid-line parietal). ***** (Electrodes with *p* < 0.05).

**Figure 3 behavsci-09-00023-f003:**
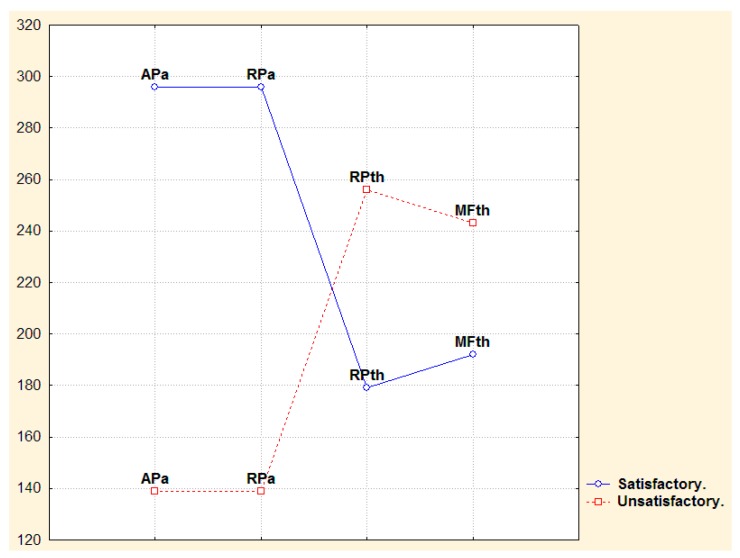
Mann Whitney. Representation of mirror behavior of theta and alpha broad-band measures in satisfactory and unsatisfactory post-surgical evolution patients in the second year after surgery. ThrP (theta relative power), ThMF (theta mean frequency), AaP (alpha absolute power), ArP (alpha relative power).

**Figure 4 behavsci-09-00023-f004:**
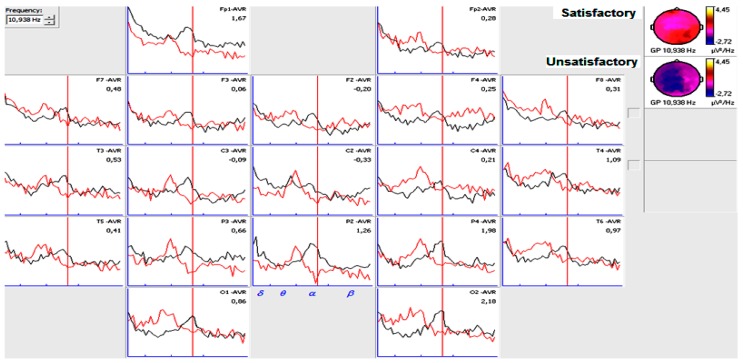
Band quantitative analysis in patients six months after surgery with epilepsy in the right temporal lobe and different evolution in the second year. Black spectrum (Frequency composition of a patient six months after surgery and satisfactory post-surgical evolution in the second year). Red spectrum (Frequency composition of a patient six months after surgery and unsatisfactory post-surgical evolution in the second year).

**Figure 5 behavsci-09-00023-f005:**
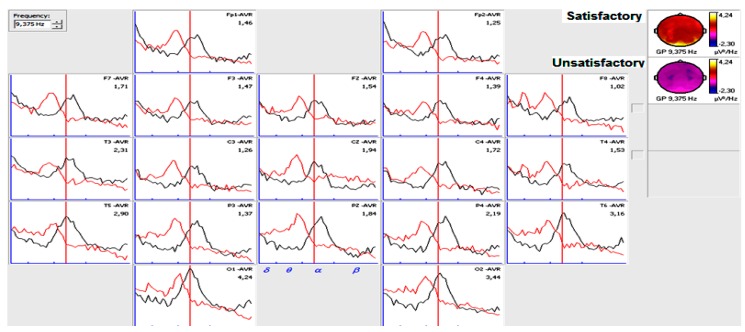
Band quantitative analysis in patients six months after surgery with epilepsy in the left temporal lobe and different evolution in the second year.Black spectrum (Frequency composition of a patient six months after surgery and satisfactory post-surgical evolution in the second year). Red spectrum (Frequency composition of a patient six months after surgery and unsatisfactory post-surgical evolution in the second year).

**Table 1 behavsci-09-00023-t001:** Ageat surgery, gender, medical history, age of first seizure, epilepsy duration, histopathology, related temporal lobe to epileptogenic zone, Engels classification. ME (meningo-encephalitis), FS (febrile seizure), N (None), PI (perinatal incident), CCT (cranial cephalic trauma), SR (speech retardation). FCD (focal cortical dysplasia) CD (chronic damage), DP (dual pathology).

Patient	Age at Surgery	Gender	Medical History	First Seizure Age	Epilepsy Duration	Histopathology	Affected Temporal Lobe	Engels Classification
1	39	m	ME	29	10	FCD IIIa	Right	IIA
2	31	m	FS	1	30	FCD IIIa	Left	IA
3	33	m	ME	6	27	FCD IIIa	Left	IA
4	41	f	N	12	29	CD	Left	IA
5	37	f	N	29	8	DP	Left	IA
6	35	f	FS	1/5	35	FCD IIIa	Right	IA
7	29	f	FS	1	28	DP	Right	IIIA
8	52	m	N	27	25	CD	Left	IA
9	26	f	FS	1/2	26	FCD IIIa	Left	IIA
10	41	f	N	26	15	FCD IIIa	Left	IIIA
11	38	f	PI	20	18	FCD IIIa	Left	IIIA
12	26	f	CCT	15	11	FCD IIIa	Right	IIA
13	34	f	FS	5/7	34	FCD IIIa	Left	IA
14	36	m	ME	23	13	CD	Left	IVA
15	23	f	N	7	16	FCD IIIa	Left	IIB
16	35	m	ME	1	34	FCD IIIa	Right	IA
17	32	f	N	22	10	FCD IIIa	Right	IA
18	37	f	N	16	21	CD	Right	IA
19	29	m	CCT	18	12	FCD IIIa	Right	IIIA
20	21	m	SR	19	2	FCD IIIb	Left	IA
21	35	f	PI	14	21	FCD IIIb	Left	IA
22	32	f	FS	13	19	FCD IIIb	Left	IIA
23	25	f	ME	0.09	25	FCD IIIa	Left	IA
24	43	m	FS	1	42	FCD IIIa	Right	IIA
25	38	m	N	10	38	FCD IIIa	Right	IA
26	37	m	FS	10	27	FCD IIIa	Right	IIB
27	54	m	N	20	15	FCD IIIa	Right	IA
28	32	m	N	18	14	FCD IIIc	Right	IIA
29	26	f	N	5	42	FCD IIIa	Left	IA

**Table 2 behavsci-09-00023-t002:** EEG spectral frequency changes from pre-surgical to post-surgical phase. Wilcoxon test. Mean comparison, µv2Hz (Microvolts square per Hertz).

**Alpha Absolute Power µv2 Hz**	**Pre-surgical**	**Post-surgical**	**Significance Electrodes**	**p-Value** **Wilcoxon Test**	**Power Augmentation In Second Year**	**p-Value** **Mean comparison**
Satisfactory post-surgical evolution in the second year	6	9.6	Frontal inferior ipsilateral	0.017	12.1	0.001
	5.2	13.7	Temporal anterior ipsilateral	0.017		
Unsatisfactory post-surgical evolution in the second year	7.3	12	Temporal anterior ipsilateral	0.010	6.1	
	5.7	7.1	Central ipsilateral	0.028		
**Theta Absolute Power µv2Hz**	**Pre-surgical**	**Post-surgical**	**Significance Electrodes**	**p-Value** **Wilcoxon Test**	**Power Augmentation In Second Year**	**p-Value** **Mean Comparison**
Satisfactory post-surgical evolution in the second year	5.9	14.8	Temporal anterior ipsilateral	0.042	8.9	0.700
Unsatisfactory post-surgical evolution in the second year	9.5	17.6	Temporal anterior ipsilateral	0.007	8.1	

**Table 3 behavsci-09-00023-t003:** Predictive analysis of frequency value with maximum energy in occipital regions in the first six months after surgery for two years seizure recurrence in patients with pharmacoresistant temporal lobe epilepsy. Logit Regression. µv2Hz (Square microvolt per Hertz).

Narrow-Band µv2Hz	Post-Surgical Evolution in the Second Year	Predicted		Percent Correct	Odds Ratio	P Value
		Satisfactory	Unsatisfactory			
Frequency value with maximum energy in occipital regions in the first six months	Satisfactory	13	3	81.25	4.333	0.04
	Unsatisfactory	6	6	50.0		

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
