# Peer review of "Association Between Quantitative Electroencephalogram Frequency Composition and Post-Surgical Evolution in Pharmacoresistant Temporal Lobe Epilepsy Patients"

_behavsci, 2019, doi:10.3390/bs9030023_

Round 1
Reviewer 1 Report
In their manuscript “Association between quantitative electroencephalogram frequency composition and postsurgical evolution in Pharmacoresistance 4 temporal lobe epilepsy patients.“ Sedeño and colleagues examined the relationship between theta and alpha power in scalp EEG and postsurgical outcome of epilepsy surgery at one year. To this extent they examined alpha and theta power in EEG of patients at 6 month and 2 years after surgery. They state that the most important finding of the study was that QEEGC in the second year was between subjects with satisfactory and unsatisfactory postsurgical outcome. In addition they claim that QEEGC at 6 months showed predictive value for two years seizure recurrence.
Here are my main concerns with this manuscript:
My biggest concern is that the statistic behind the calculations is not clear. E.g. Table 2: There are three electrode locations mentioned: Frontal inferior ipsilateral /temporal inferior ipsilateral/central ipsilateral. I am confused as to which electrodes were tested. All electrodes tested (even if not significant) should be included in the table. The authors should also state if the accounted for multiple testing (i.e. was the threshold for the p test adjusted in an post hoc test (e.g. post hoc tukey’s test) and did the authors do a test an ANOVA to test if there is a significant difference between the groups. This is very important since single testing with so many electrodes will yield significances by chance and therefore the p value has to be adjusted. Otherwise the results are meaningless.
Another problem is that the authors do not state how large the resection volumes were. Could this be a confounder? It could be that this may explain why more theta is in the non-satisfactory outcome group. It could also be that the resection volume was overall larger in the non-satisfactory group.
It would be also interesting to classify the patient according to the Engels classification and to see whether there is a correlation between the outcome as classified by the Engels classification system and the theta and alpha power. The authors could try and plot such correlations.
I do not fully understand the patient composition. I am surprised that none of the patients had “pure” HS, but that it was always associated with FCD (i.e. therefore FCD Type IIIa). This is quite unusual for an epilepsy series, where HS should be represented to some extent. Does lesion type in the table relate to histopathology? If so, the authors should write histopathology instead of lesion type in the table
Another question relates to the resection: Were the lesions fully resected or was there anything left behind and seen on MRI scan post-surgery? This could also explain the difference in frequency bands post-surgery in the different outcome groups.
Figure 1: the style of the figure should be improved. The subscription “2S and 2U” should be changed to unsatisfactory and satisfactory outcome. Also the p value should not be written next to the figure. Overall this looks like an output of a statistic calculation and not an edited figure. The values in the boxplots should be omitted.
Figure 2 is difficult to understand. I would suggest mentioning all electrodes in the figure even if they are not significant…. to keep it more logic. Perhaps one could do that in a model head.
Figure 3 A: this plot is difficult to understand. I would suggest omitting it.
In 3 B the following abbreviations are used: ThrP (theta relative power), ThMF (theta mean frequency), AaP (alpha 214 absolute power), ArP (alpha relative power). I do not understand why there is no consistency and why theta mean frequency but alpha absolute power is used. I would suggest consistently comparing the same aspects in the different frequency bands.
Table one: In the section medical conditions, I would suggest to name only conditions related to the brain and such that are risk factors for epilepsy and not such as Asthma and diabetes; The Abbreviation CET is not explained. Which pathologies were included in dual pathology? In the table the Engels classification should be mentioned in addition to the classification satisfactory and unsatisfactory
Table 3 is entirely unclear to me. What does that mean? This is not described sufficiently in the text. In the unsatisfactory raw the percent correct is 50% - this is similar to guessing and thus for me is no “predictor”. I am also uncler how the authors come to the conclusion that they are able to predict outcome of epilepsy surgery?
Figure 4 and 5: As far as I understand figure 4 and 5 relates to two patients (i.e. four patients in total); It is important that this is presented for all patients. This would then be the central figure and the central finding of the paper. In this case the frequency spectrum could be presented with a line with error bars for both satisfactory and unsatisfactory outcome.
Minor points:
The authors should have the paper revised by a native speaker. There are many sentences which are difficult to understand. E.g. “QEEGC can be a tool to know the postsurgical evolution of patients “ should be changed into:
“QEEG can be a tool to predict outcome of epilepsy surgery” or in the introduction “ The objective of this investigation was to know the behavior of QEEGC” or in the methods: “It was used Neuronic EEG Tomographic Quantitative 6.2.2.0 for Quantitative analyses of 99 narrow and broad frequency bands with eyes closed. „… this is wrong syntax. There are many more examples and I cannot name all of them. The whole manuscript needs to be revised for grammatical errors and style.
Author Response
Reviewer answers.
Abstract.
A prospective, longitudinal study was made at International Neurological Restoration Center, Havana, Cuba.
2. QEEGC can be a tool to predict outcome of epilepsy surgery.
Introduction.
3. It was added reference 2: Armonk C, Radtke RA, Friedman AH et al. Predictors of outcome of epilepsy surgery: multivariate analysis with validation. Epilepsia 1996;37: 814–21.
4. Most of the researches have been focused in predictive value of epileptiform discharges for post-surgical evolution, while some results have shown strong predictive value of bad evolution [2-9], others have not found any relation [10, 11].
Materials and Methods.
5. Subjects and EEG recordings were evaluated by two epileptologists (LM and RV) before surgery, and six months and two years after.
6. Patients older than 18 years were included after their written consent to participate in the study. Electroencephalograms were registered before surgery, and six month and two years after.
Results.
7. The difference between satisfactory and unsatisfactory postsurgical evolution groups was not statistically significant.
8. Table 1. What we wanted to mean: Age at surgery.
9. Figure 1. What we wanted to mean: Patients in pre-surgical phase; regroup according to the evolution achieved in the second year of surgical intervention.
Reviewer answers.
P values: When examining results exact probabilities computes in Statistica program for small to moderate sized samples were taken.
Resection volumes: When examining resection volumes, there were no significant differences in the resection size between right and left temporal lobectomies (Mann Whitney test p>0.05).
We did not applied correlations between Engels classification and alpha/theta power because Engels scale is a category variable that goes from I to IV. Also our sample is small.
Lesion type in the table does relate to histopathology. Hippocampal sclerosis and Focal cortical dysplasia were confirmed by histopathology. Cases with FCD type IIIA had Hippocampal sclerosis plus Mild Focal Cortical Dysplasia. Dual pathology was related to pylocitic astrocytom and arachnoid cystic in two of the patients. Morales Chacón L M, Garcia Maeso I, Báez Martin M M, Bender del Busto J E, García Navarro M E, Quintanal Cordero N et al. Long-Term Electroclinical and Employment Follow up in Temporal Lobe Epilepsy Surgery. A Cuban Comprehensive Epilepsy Surgery Program. Article. Behavioral sciences. Behav. Sci. 2018, 8, 19. www.mdpi.com/journal/behavsci.
Where lesions fully resected? The resection was made tailored by sequential intraoperative Electrocorticography until epileptiform activity disappeared.
Was there anything left and seen of imaging test during postsurgical phase: It could be a factor that explains some changes, further investigations are needed.
Figure 2 was changed. Four measures are now display differently and separately. All electrodes are shown.
8. Figure 3b: We did not make ourselves clear: Figure 3 is a representation of dissimilar findings of alpha and theta frequency measures in patients with satisfactory and unsatisfactory post-surgical evolution in the second year. Significance measures are represented. It can be seen as part of figure 2. It can be observed a mirror effect: while satisfactory postsurgical evolution patients in the second year showed higher power of alpha measures, unsatisfactory postsurgical evolution patients showed higher power of theta measures. Same aspects were compared between groups.
Table one (medical conditions): CET is referred to CCT (Cranial cephalic trauma), there was a mistake there. Engels classification was included. Asthma and diabetes are not named.
Reviewer answers.
Abstract.
A prospective, longitudinal study was made at International Neurological Restoration Center, Havana, Cuba.
2. QEEGC can be a tool to predict outcome of epilepsy surgery.
Introduction.
3. It was added reference 2: Armonk C, Radtke RA, Friedman AH et al. Predictors of outcome of epilepsy surgery: multivariate analysis with validation. Epilepsia 1996;37: 814–21.
4. Most of the researches have been focused in predictive value of epileptiform discharges for post-surgical evolution, while some results have shown strong predictive value of bad evolution [2-9], others have not found any relation [10, 11].
Materials and Methods.
5. Subjects and EEG recordings were evaluated by two epileptologists (LM and RV) before surgery, and six months and two years after.
6. Patients older than 18 years were included after their written consent to participate in the study. Electroencephalograms were registered before surgery, and six month and two years after.
Results.
7. The difference between satisfactory and unsatisfactory postsurgical evolution groups was not statistically significant.
8. Table 1. What we wanted to mean: Age at surgery.
9. Figure 1. What we wanted to mean: Patients in pre-surgical phase; regroup according to the evolution achieved in the second year of surgical intervention.
Reviewer answers.
P values: When examining results exact probabilities computes in Statistica program for small to moderate sized samples were taken.
Resection volumes: When examining resection volumes, there were no significant differences in the resection size between right and left temporal lobectomies (Mann Whitney test p>0.05).
We did not applied correlations between Engels classification and alpha/theta power because Engels scale is a category variable that goes from I to IV. Also our sample is small.
Lesion type in the table does relate to histopathology. Hippocampal sclerosis and Focal cortical dysplasia were confirmed by histopathology. Cases with FCD type IIIA had Hippocampal sclerosis plus Mild Focal Cortical Dysplasia. Dual pathology was related to pylocitic astrocytom and arachnoid cystic in two of the patients. Morales Chacón L M, Garcia Maeso I, Báez Martin M M, Bender del Busto J E, García Navarro M E, Quintanal Cordero N et al. Long-Term Electroclinical and Employment Follow up in Temporal Lobe Epilepsy Surgery. A Cuban Comprehensive Epilepsy Surgery Program. Article. Behavioral sciences. Behav. Sci. 2018, 8, 19. www.mdpi.com/journal/behavsci.
Where lesions fully resected? The resection was made tailored by sequential intraoperative Electrocorticography until epileptiform activity disappeared.
Was there anything left and seen of imaging test during postsurgical phase: It could be a factor that explains some changes, further investigations are needed.
Figure 2 was changed. Four measures are now display differently and separately. All electrodes are shown.
8. Figure 3b: We did not make ourselves clear: Figure 3 is a representation of dissimilar findings of alpha and theta frequency measures in patients with satisfactory and unsatisfactory post-surgical evolution in the second year. Significance measures are represented. It can be seen as part of figure 2. It can be observed a mirror effect: while satisfactory postsurgical evolution patients in the second year showed higher power of alpha measures, unsatisfactory postsurgical evolution patients showed higher power of theta measures. Same aspects were compared between groups.
Table one (medical conditions): CET is referred to CCT (Cranial cephalic trauma), there was a mistake there. Engels classification was included. Asthma and diabetes are not named.
Reviewer answers.
Abstract.
A prospective, longitudinal study was made at International Neurological Restoration Center, Havana, Cuba.
2. QEEGC can be a tool to predict outcome of epilepsy surgery.
Introduction.
3. It was added reference 2: Armonk C, Radtke RA, Friedman AH et al. Predictors of outcome of epilepsy surgery: multivariate analysis with validation. Epilepsia 1996;37: 814–21.
4. Most of the researches have been focused in predictive value of epileptiform discharges for post-surgical evolution, while some results have shown strong predictive value of bad evolution [2-9], others have not found any relation [10, 11].
Materials and Methods.
5. Subjects and EEG recordings were evaluated by two epileptologists (LM and RV) before surgery, and six months and two years after.
6. Patients older than 18 years were included after their written consent to participate in the study. Electroencephalograms were registered before surgery, and six month and two years after.
Results.
7. The difference between satisfactory and unsatisfactory postsurgical evolution groups was not statistically significant.
8. Table 1. What we wanted to mean: Age at surgery.
9. Figure 1. What we wanted to mean: Patients in pre-surgical phase; regroup according to the evolution achieved in the second year of surgical intervention.
Reviewer answers.
P values: When examining results exact probabilities computes in Statistica program for small to moderate sized samples were taken.
Resection volumes: When examining resection volumes, there were no significant differences in the resection size between right and left temporal lobectomies (Mann Whitney test p>0.05).
We did not applied correlations between Engels classification and alpha/theta power because Engels scale is a category variable that goes from I to IV. Also our sample is small.
Lesion type in the table does relate to histopathology. Hippocampal sclerosis and Focal cortical dysplasia were confirmed by histopathology. Cases with FCD type IIIA had Hippocampal sclerosis plus Mild Focal Cortical Dysplasia. Dual pathology was related to pylocitic astrocytom and arachnoid cystic in two of the patients. Morales Chacón L M, Garcia Maeso I, Báez Martin M M, Bender del Busto J E, García Navarro M E, Quintanal Cordero N et al. Long-Term Electroclinical and Employment Follow up in Temporal Lobe Epilepsy Surgery. A Cuban Comprehensive Epilepsy Surgery Program. Article. Behavioral sciences. Behav. Sci. 2018, 8, 19. www.mdpi.com/journal/behavsci.
Where lesions fully resected? The resection was made tailored by sequential intraoperative Electrocorticography until epileptiform activity disappeared.
Was there anything left and seen of imaging test during postsurgical phase: It could be a factor that explains some changes, further investigations are needed.
Figure 2 was changed. Four measures are now display differently and separately. All electrodes are shown.
8. Figure 3b: We did not make ourselves clear: Figure 3 is a representation of dissimilar findings of alpha and theta frequency measures in patients with satisfactory and unsatisfactory post-surgical evolution in the second year. Significance measures are represented. It can be seen as part of figure 2. It can be observed a mirror effect: while satisfactory postsurgical evolution patients in the second year showed higher power of alpha measures, unsatisfactory postsurgical evolution patients showed higher power of theta measures. Same aspects were compared between groups.
Table one (medical conditions): CET is referred to CCT (Cranial cephalic trauma), there was a mistake there. Engels classification was included. Asthma and diabetes are not named.
Reviewer answers.
Abstract.
A prospective, longitudinal study was made at International Neurological Restoration Center, Havana, Cuba.
2. QEEGC can be a tool to predict outcome of epilepsy surgery.
Introduction.
3. It was added reference 2: Armonk C, Radtke RA, Friedman AH et al. Predictors of outcome of epilepsy surgery: multivariate analysis with validation. Epilepsia 1996;37: 814–21.
4. Most of the researches have been focused in predictive value of epileptiform discharges for post-surgical evolution, while some results have shown strong predictive value of bad evolution [2-9], others have not found any relation [10, 11].
Materials and Methods.
5. Subjects and EEG recordings were evaluated by two epileptologists (LM and RV) before surgery, and six months and two years after.
6. Patients older than 18 years were included after their written consent to participate in the study. Electroencephalograms were registered before surgery, and six month and two years after.
Results.
7. The difference between satisfactory and unsatisfactory postsurgical evolution groups was not statistically significant.
8. Table 1. What we wanted to mean: Age at surgery.
9. Figure 1. What we wanted to mean: Patients in pre-surgical phase; regroup according to the evolution achieved in the second year of surgical intervention.
Reviewer answers.
P values: When examining results exact probabilities computes in Statistica program for small to moderate sized samples were taken.
Resection volumes: When examining resection volumes, there were no significant differences in the resection size between right and left temporal lobectomies (Mann Whitney test p>0.05).
We did not applied correlations between Engels classification and alpha/theta power because Engels scale is a category variable that goes from I to IV. Also our sample is small.
Lesion type in the table does relate to histopathology. Hippocampal sclerosis and Focal cortical dysplasia were confirmed by histopathology. Cases with FCD type IIIA had Hippocampal sclerosis plus Mild Focal Cortical Dysplasia. Dual pathology was related to pylocitic astrocytom and arachnoid cystic in two of the patients. Morales Chacón L M, Garcia Maeso I, Báez Martin M M, Bender del Busto J E, García Navarro M E, Quintanal Cordero N et al. Long-Term Electroclinical and Employment Follow up in Temporal Lobe Epilepsy Surgery. A Cuban Comprehensive Epilepsy Surgery Program. Article. Behavioral sciences. Behav. Sci. 2018, 8, 19. www.mdpi.com/journal/behavsci.
Where lesions fully resected? The resection was made tailored by sequential intraoperative Electrocorticography until epileptiform activity disappeared.
Was there anything left and seen of imaging test during postsurgical phase: It could be a factor that explains some changes, further investigations are needed.
Figure 2 was changed. Four measures are now display differently and separately. All electrodes are shown.
8. Figure 3b: We did not make ourselves clear: Figure 3 is a representation of dissimilar findings of alpha and theta frequency measures in patients with satisfactory and unsatisfactory post-surgical evolution in the second year. Significance measures are represented. It can be seen as part of figure 2. It can be observed a mirror effect: while satisfactory postsurgical evolution patients in the second year showed higher power of alpha measures, unsatisfactory postsurgical evolution patients showed higher power of theta measures. Same aspects were compared between groups.
Table one (medical conditions): CET is referred to CCT (Cranial cephalic trauma), there was a mistake there. Engels classification was included. Asthma and diabetes are not named.
Table 1. Age at surgery, gender, medical history, age of first seizure, epilepsy duration, histopathology, related temporal lobe to epileptogenic zone, Engels classification. ME (meningo-encephalitis), FS (febrile seizure), N (None), PI (perinatal incident), CCT (cranial cephalic trauma), SR (speech retardation). FCD (focal cortical dysplasia) CD (chronic damage), DP (dual pathology).
Patients. | Age at surgery. | Gender. | Medical history. | First seizure Age. | Epilepsy duration. | Histopathology. | Affected temporal lobe. | Engels classification. |
1 | 39 | m | ME | 29 | 10 | FCD IIIa | Right. | IIA |
2 | 31 | m | FS | 1 | 30 | FCD IIIa | Left. | IA |
3 | 33 | m | ME | 6 | 27 | FCD IIIa | Left. | IA |
4 | 41 | f | N | 12 | 29 | CD | Left. | IA |
5 | 37 | f | N | 29 | 8 | DP | Left. | IA |
6 | 35 | f | FS | 1/5 | 35 | FCD IIIa | Right. | IA |
7 | 29 | f | FS | 1 | 28 | DP | Right | IIIA |
8 | 52 | m | N | 27 | 25 | CD | Left. | IA |
9 | 26 | f | FS | 1/2 | 26 | FCD IIIa | Left. | IIA |
10 | 41 | f | N | 26 | 15 | FCD IIIa | Left. | IIIA |
11 | 38 | f | PI | 20 | 18 | FCD IIIa | Left. | IIIA |
12 | 26 | f | CCT | 15 | 11 | FCD IIIa | Right. | IIA |
13 | 34 | f | FS | 5/7 | 34 | FCD IIIa | Left. | IA |
14 | 36 | m | ME | 23 | 13 | CD | Left. | IVA |
15 | 23 | f | N | 7 | 16 | FCD IIIa | Left. | IIB |
16 | 35 | m | ME | 1 | 34 | FCD IIIa | Right | IA |
17 | 32 | f | N | 22 | 10 | FCD IIIa | Right | IA |
18 | 37 | f | N | 16 | 21 | CD | Right | IA |
19 | 29 | m | CCT | 18 | 12 | FCD IIIa | Right. | IIIA |
20 | 21 | m | SR | 19 | 2 | FCD IIIb | Left. | IA |
21 | 35 | f | PI | 14 | 21 | FCD IIIb | Left. | IA |
22 | 32 | f | FS | 13 | 19 | FCD IIIb | Left. | IIA |
23 | 25 | f | ME | 0,09 | 25 | FCD IIIa | Left. | IA |
24 | 43 | m | FS | 1 | 42 | FCD IIIa | Right. | IIA |
25 | 38 | m | N | 10 | 38 | FCD IIIa | Right | IA |
26 | 37 | m | FS | 10 | 27 | FCD IIIa | Right. | IIB |
27 | 54 | m | N | 20 | 15 | FCD IIIa | Right | IA |
28 | 32 | m | N | 18 | 14 | FCD IIIc | Right. | IIA |
29 | 26 | f | N | 5 | 42 | FCD IIIa | Left. | IA |
1. Table 3 shows the modeled probability that evolution is equal to satisfactory. We are aware that for evolution prediction the percent of correct prediction of unsatisfactory group is meaningless. Nevertheless we observed that for both groups theta absolute power augmented from presurgical to postsurgical phase, but when comparing this increment between groups, it was not statistically different. On the other hand alpha absolute power augmented in both groups from presurgical to postsurgical phase but more in satisfactory group, even with significance differences between groups. So differences in absolute power after surgery were mainly related to alpha band. Taking into account this findings and the significance of the Logit test, it could indicate that alpha power increment after surgery could be more precise for evolution prediction than theta augmentation; however we know that for a small sample, further studies are needed.
Table 3: Total of correct and incorrect prediction of cases for each group is showed, satisfactory (13 correct and 3 incorrect) and unsatisfactory (6 correct and 6 incorrect). While satisfactory evolution in the second year showed 81, 25% of correct prediction, unsatisfactory evolution was only 50% correct.
Figures 4 and 5 relates to two patients (four in total) because our quantitative analysis program is not able to do an average spectrum for each group, but only comparisons of two patients each time.
Presurgical EEG vs. postsurgical EEG. Patients regroup according to satisfactory and unsatisfactory evolution achieved in the second year. All electrodes are shown in the table.
AaP (µv2 Hz) | Presurgical vs Postsurgical phase. Electrodes. Wilcoxon p value. | ||||||||||||||||||
Satisfactory. | Fpi 0,49 | Fpc 0,49 | Fsi 0,73 | Fsc 0,31 | Ci 0,17 | Cc 0,73 | Pi 0,75 | Pc 0,73 | Oi 0,49 | Oc 0,39 | Fii 0,01 | Fic 0,86 | Tai 0,01 | Tac 0,73 | Tpi 0,49 | Tpc 0,73 | Fz 0,39 | Cz 0,39 | Pz 0,86 |
Unsatisfactory. | Fpi 0,44 | Fpc 0,51 | Fsi 0,37 | Fsc 0,06 | Ci 0,02 | Cc 0,26 | Pi 0,44 | Pc 0,20 | Oi 0,59 | Oc 0,59 | Fii 0,95 | Fic 0,37 | Tai 0,01 | Tac 0,76 | Tpi 0,51 | Tpc 0,26 | Fz 0,67 | Cz 0,95 | Pz 0,066 |
Thap (µv2 Hz) | Presurgical vs Postsurgical phase. Electrodes. Wilcoxon p value. | ||||||||||||||||||
Satisfactory. | Fpi 0,86 | Fpc 1,00 | Fsi 0,44 | Fsc 0,61 | Ci 0,31 | Cc 0,86 | Pi 0,86 | Pc 0,73 | Oi 0,73 | Oc 0,86 | Fii 0,39 | Fic 0,49 | Tai 0,04 | Tac 0,75 | Tpi 0,39 | Tpc 0,86 | Fz 0,31 | Cz 0,12 | Pz 0,39 |
Unsatisfactory. | Fpi 0,67 | Fpc 0,31 | Fsi 0,31 | Fsc 0,59 | Ci 0,13 | Cc 0,31 | Pi 0,76 | Pc 0,95 | Oi 0,44 | Oc 0,51 | Fii 0,95 | Fic 0,37 | Tai 0,00 | Tac 0,26 | Tpi 0,76 | Tpc 0,51 | Fz 0,76 | Cz 0,31 | Pz 0,59 |

Reviewer 2 Report
Review of the manuscript “Association between quantitative electroencephalogram frequency composition and postsurgical evolution in Pharmacoresistance temporal lobe epilepsy patients” by Raúl Roberto Valdés Sedeño and coauthors submitted for consideration to “Biomolecules”.
The authors of the manuscript put forward two aims in a prospective, longitudinal study: Aim 1). To investigate the association between quantitative electroencephalogram frequency composition (QEEGC) and postsurgical evolution in patients with Pharmacoresistance Temporal Lobe Epilepsy (TLE) and Aim 2). To evaluate the predictive value of QEEGC before and after the surgery.
Predictive value of epileptiform discharges for post-surgical evolution is currently a contradictory issue requiring more detailed and systematic examination. In spite of many attempts to clarify the reliability of epileptiform discharges by several research teams, there is still no agreement concerning biomarkers of post-surgical evolution.
The authors make an interesting conclusion that QEEGC is different between subjects with satisfactory and unsatisfactory postsurgical evolution in the second year, and that six month QEEGC showed predictive value for two years seizure recurrence. Therefore, the topic of the manuscript is important and the results will be interesting for the readership of “Biomolecules”. However, the manuscript contains drawbacks mentioned below.
The following corrections should be made:
Abstract
1 Lines 15-16 “A prospective, longitudinal study was made at International Neurological Restoration Center”
More details should be added, for example, Havana, Cuba.
2 Lines 27-28 “QEEGC was different between unsatisfactory and satisfactory postsurgical clinical evolution patients, six month after surgery QEEGC can be a tool to know the postsurgical evolution of patients at two years”.
The authors should not combine a sentence with their results and conclusion. The last sentence in the Abstract should briefly summarize the main outcome of the investigation.
Introduction
Line 34 “Several authors have described the prognostic value…” If the authors mention several authors, the need to give at least two references.
Line 36 “Predictive value of epileptiform discharges for post-surgical evolution have been analyzed mainly, …” The sense of this sentence is not clear. What “mainly” stands for here? The sentence should be rewritten in a more clear way.
Materials and Methods
Line 57. “Patients and neurophysiology studies were evaluated by epileptologists (LM) an (RV) before surgery”. This is an awkward sentence which should be rewritten.
Line 65 “Patients that gave their consent to participate. Being 18 years or older. Electroencephalogram registration before surgery, six month and two years after [20].”
These sentences should be combined and written clearly, for example:” Patients older than 18 years were included after their written consent to participate in the study. Electroencephalograms were registered before surgery, and six month and two years after [20].
Results
Line 121 “Comparisons between satisfactory and unsatisfactory postsurgical evolution groups was not statistically different.”
This is an awkward sentence, which should be corrected, for example: “The difference between satisfactory and unsatisfactory postsurgical evolution groups was not statistically significant”.
Table 1.
Line 125. Why the text in the legend begins with “at” ?.
Figure 1, Line 139. Why the legend begins with “. of patients in pre-surgical phase; regroup according to the evolution achieved in the second
Author Response
Reviewer answers.
Abstract.
A prospective, longitudinal study was made at International Neurological Restoration Center, Havana, Cuba.
2. QEEGC can be a tool to predict outcome of epilepsy surgery.
Introduction.
3. It was added reference 2: Armonk C, Radtke RA, Friedman AH et al. Predictors of outcome of epilepsy surgery: multivariate analysis with validation. Epilepsia 1996;37: 814–21.
4. Most of the researches have been focused in predictive value of epileptiform discharges for post-surgical evolution, while some results have shown strong predictive value of bad evolution [2-9], others have not found any relation [10, 11].
Materials and Methods.
5. Subjects and EEG recordings were evaluated by two epileptologists (LM and RV) before surgery, and six months and two years after.
6. Patients older than 18 years were included after their written consent to participate in the study. Electroencephalograms were registered before surgery, and six month and two years after.
Results.
7. The difference between satisfactory and unsatisfactory postsurgical evolution groups was not statistically significant.
8. Table 1. What we wanted to mean: Age at surgery.
9. Figure 1. What we wanted to mean: Patients in pre-surgical phase; regroup according to the evolution achieved in the second year of surgical intervention.
Reviewer answers.
P values: When examining results exact probabilities computes in Statistica program for small to moderate sized samples were taken.
Resection volumes: When examining resection volumes, there were no significant differences in the resection size between right and left temporal lobectomies (Mann Whitney test p>0.05).
We did not applied correlations between Engels classification and alpha/theta power because Engels scale is a category variable that goes from I to IV. Also our sample is small.
Lesion type in the table does relate to histopathology. Hippocampal sclerosis and Focal cortical dysplasia were confirmed by histopathology. Cases with FCD type IIIA had Hippocampal sclerosis plus Mild Focal Cortical Dysplasia. Dual pathology was related to pylocitic astrocytom and arachnoid cystic in two of the patients. Morales Chacón L M, Garcia Maeso I, Báez Martin M M, Bender del Busto J E, García Navarro M E, Quintanal Cordero N et al. Long-Term Electroclinical and Employment Follow up in Temporal Lobe Epilepsy Surgery. A Cuban Comprehensive Epilepsy Surgery Program. Article. Behavioral sciences. Behav. Sci. 2018, 8, 19. www.mdpi.com/journal/behavsci.
Where lesions fully resected? The resection was made tailored by sequential intraoperative Electrocorticography until epileptiform activity disappeared.
Was there anything left and seen of imaging test during postsurgical phase: It could be a factor that explains some changes, further investigations are needed.
Figure 2 was changed. Four measures are now display differently and separately. All electrodes are shown.
8. Figure 3b: We did not make ourselves clear: Figure 3 is a representation of dissimilar findings of alpha and theta frequency measures in patients with satisfactory and unsatisfactory post-surgical evolution in the second year. Significance measures are represented. It can be seen as part of figure 2. It can be observed a mirror effect: while satisfactory postsurgical evolution patients in the second year showed higher power of alpha measures, unsatisfactory postsurgical evolution patients showed higher power of theta measures. Same aspects were compared between groups.
Table one (medical conditions): CET is referred to CCT (Cranial cephalic trauma), there was a mistake there. Engels classification was included. Asthma and diabetes are not named.
Table 1. Age at surgery, gender, medical history, age of first seizure, epilepsy duration, histopathology, related temporal lobe to epileptogenic zone, Engels classification. ME (meningo-encephalitis), FS (febrile seizure), N (None), PI (perinatal incident), CCT (cranial cephalic trauma), SR (speech retardation). FCD (focal cortical dysplasia) CD (chronic damage), DP (dual pathology).
Patients. | Age at surgery. | Gender. | Medical history. | First seizure Age. | Epilepsy duration. | Histopathology. | Affected temporal lobe. | Engels classification. |
1 | 39 | m | ME | 29 | 10 | FCD IIIa | Right. | IIA |
2 | 31 | m | FS | 1 | 30 | FCD IIIa | Left. | IA |
3 | 33 | m | ME | 6 | 27 | FCD IIIa | Left. | IA |
4 | 41 | f | N | 12 | 29 | CD | Left. | IA |
5 | 37 | f | N | 29 | 8 | DP | Left. | IA |
6 | 35 | f | FS | 1/5 | 35 | FCD IIIa | Right. | IA |
7 | 29 | f | FS | 1 | 28 | DP | Right | IIIA |
8 | 52 | m | N | 27 | 25 | CD | Left. | IA |
9 | 26 | f | FS | 1/2 | 26 | FCD IIIa | Left. | IIA |
10 | 41 | f | N | 26 | 15 | FCD IIIa | Left. | IIIA |
11 | 38 | f | PI | 20 | 18 | FCD IIIa | Left. | IIIA |
12 | 26 | f | CCT | 15 | 11 | FCD IIIa | Right. | IIA |
13 | 34 | f | FS | 5/7 | 34 | FCD IIIa | Left. | IA |
14 | 36 | m | ME | 23 | 13 | CD | Left. | IVA |
15 | 23 | f | N | 7 | 16 | FCD IIIa | Left. | IIB |
16 | 35 | m | ME | 1 | 34 | FCD IIIa | Right | IA |
17 | 32 | f | N | 22 | 10 | FCD IIIa | Right | IA |
18 | 37 | f | N | 16 | 21 | CD | Right | IA |
19 | 29 | m | CCT | 18 | 12 | FCD IIIa | Right. | IIIA |
20 | 21 | m | SR | 19 | 2 | FCD IIIb | Left. | IA |
21 | 35 | f | PI | 14 | 21 | FCD IIIb | Left. | IA |
22 | 32 | f | FS | 13 | 19 | FCD IIIb | Left. | IIA |
23 | 25 | f | ME | 0,09 | 25 | FCD IIIa | Left. | IA |
24 | 43 | m | FS | 1 | 42 | FCD IIIa | Right. | IIA |
25 | 38 | m | N | 10 | 38 | FCD IIIa | Right | IA |
26 | 37 | m | FS | 10 | 27 | FCD IIIa | Right. | IIB |
27 | 54 | m | N | 20 | 15 | FCD IIIa | Right | IA |
28 | 32 | m | N | 18 | 14 | FCD IIIc | Right. | IIA |
29 | 26 | f | N | 5 | 42 | FCD IIIa | Left. | IA |
1. Table 3 shows the modeled probability that evolution is equal to satisfactory. We are aware that for evolution prediction the percent of correct prediction of unsatisfactory group is meaningless. Nevertheless we observed that for both groups theta absolute power augmented from presurgical to postsurgical phase, but when comparing this increment between groups, it was not statistically different. On the other hand alpha absolute power augmented in both groups from presurgical to postsurgical phase but more in satisfactory group, even with significance differences between groups. So differences in absolute power after surgery were mainly related to alpha band. Taking into account this findings and the significance of the Logit test, it could indicate that alpha power increment after surgery could be more precise for evolution prediction than theta augmentation; however we know that for a small sample, further studies are needed.
Table 3: Total of correct and incorrect prediction of cases for each group is showed, satisfactory (13 correct and 3 incorrect) and unsatisfactory (6 correct and 6 incorrect). While satisfactory evolution in the second year showed 81, 25% of correct prediction, unsatisfactory evolution was only 50% correct.
Figures 4 and 5 relates to two patients (four in total) because our quantitative analysis program is not able to do an average spectrum for each group, but only comparisons of two patients each time.
Presurgical EEG vs. postsurgical EEG. Patients regroup according to satisfactory and unsatisfactory evolution achieved in the second year. All electrodes are shown in the table.
AaP (µv2 Hz) | Presurgical vs Postsurgical phase. Electrodes. Wilcoxon p value. | ||||||||||||||||||
Satisfactory. | Fpi 0,49 | Fpc 0,49 | Fsi 0,73 | Fsc 0,31 | Ci 0,17 | Cc 0,73 | Pi 0,75 | Pc 0,73 | Oi 0,49 | Oc 0,39 | Fii 0,01 | Fic 0,86 | Tai 0,01 | Tac 0,73 | Tpi 0,49 | Tpc 0,73 | Fz 0,39 | Cz 0,39 | Pz 0,86 |
Unsatisfactory. | Fpi 0,44 | Fpc 0,51 | Fsi 0,37 | Fsc 0,06 | Ci 0,02 | Cc 0,26 | Pi 0,44 | Pc 0,20 | Oi 0,59 | Oc 0,59 | Fii 0,95 | Fic 0,37 | Tai 0,01 | Tac 0,76 | Tpi 0,51 | Tpc 0,26 | Fz 0,67 | Cz 0,95 | Pz 0,066 |
Thap (µv2 Hz) | Presurgical vs Postsurgical phase. Electrodes. Wilcoxon p value. | ||||||||||||||||||
Satisfactory. | Fpi 0,86 | Fpc 1,00 | Fsi 0,44 | Fsc 0,61 | Ci 0,31 | Cc 0,86 | Pi 0,86 | Pc 0,73 | Oi 0,73 | Oc 0,86 | Fii 0,39 | Fic 0,49 | Tai 0,04 | Tac 0,75 | Tpi 0,39 | Tpc 0,86 | Fz 0,31 | Cz 0,12 | Pz 0,39 |
Unsatisfactory. | Fpi 0,67 | Fpc 0,31 | Fsi 0,31 | Fsc 0,59 | Ci 0,13 | Cc 0,31 | Pi 0,76 | Pc 0,95 | Oi 0,44 | Oc 0,51 | Fii 0,95 | Fic 0,37 | Tai 0,00 | Tac 0,26 | Tpi 0,76 | Tpc 0,51 | Fz 0,76 | Cz 0,31 | Pz 0,59 |

Round 2
Reviewer 2 Report
The manuscript is improved